# Association between Macronutrient and Fatty Acid Consumption and Metabolic Syndrome: A South African Taxi Driver Survey

**DOI:** 10.3390/ijerph192315452

**Published:** 2022-11-22

**Authors:** Machoene Derrick Sekgala, Maretha Opperman, Buhle Mpahleni, Zandile June-Rose Mchiza

**Affiliations:** 1School of Public Health, University of the Western Cape, Bellville 7535, South Africa; 2Human and Social Capabilities, Human Sciences Research Council, Cape Town 8000, South Africa; 3Functional Foods Research Unit, Department of Biotechnology and Consumer Science, Cape Peninsula University of Technology, Cape Town 7535, South Africa; 4Non-Communicable Diseases Research Unit, South African Medical Research Council, Tygerberg, Cape Town 7505, South Africa

**Keywords:** macronutrient intake, fatty acid intake, metabolic syndrome, substitution mode, diet, South African taxi drivers

## Abstract

We aimed to examine the association between macronutrient and fatty acid intake and metabolic syndrome (MetS) and its components in South African male mini-bus taxi drivers. One hundred and eighty-five (*n* = 185) male taxi drivers, aged 20 years and older, who operate in the Cape Town metropole, South Africa, were included. The International Diabetes Federation (IDF) algorithm was used to define MetS. The association between macronutrient and fatty acid intake (assessed using 24 h recall) and MetS were analyzed using multivariable nutrient density substitution models. Overall, protein consumption significantly increased the likelihood of high blood pressure (HBP) and significantly lowered the likelihood of having low levels of high-density lipoprotein cholesterol (HDL-C). In an isoenergetic state, the intake of protein instead of carbohydrates (CHOs) and total fat, reduced the likelihood of elevated triglycerides by 6.7% and 6.6%, respectively. The intake of CHOs instead of protein and total fat, reduced the likelihood of HBP by 2.2% and 2.8%, respectively. In the same isoenergetic state, the intake of saturated fatty acids (SFAs) instead of mono-unsaturated fatty acids (MUFAs) increased the likelihood of HBP by 9.8%, whereas the intake of polyunsaturated fatty acids (PUFAs) instead of SFAs decreased the likelihood of HBP by 9.4%. The current study showed that when total food energy intake is kept constant, a diet that is high in protein, CHOs and PUFAs reduces triglycerides and BP, whereas the intake of total fat and SFAs had the opposite effect. It should, however, be noted that these outcomes were produced using mathematical models, as such we recommend further prospective studies in real life that will reveal the actual associations between the consumption of macronutrients and fatty acids and MetS and its components.

## 1. Introduction

Metabolic syndrome (MetS) is defined as a cluster of metabolic disorders, which include central obesity, abnormal blood glucose levels, dyslipidemia and hypertension [1,2]. According to Sekgala et al. (2018) [3], the current prevalence of MetS in adult South African men is more than 23% and is even higher among male mini-bus taxi drivers [4]. Metabolic syndrome has also been linked to an increased risk of cardiovascular disease (CVD), diabetes and chronic kidney diseases [5]. Even though the etiology of MetS is not fully understood, the interaction of genetics, lifestyle and environmental factors have been implicated in the development of MetS [6]. Among lifestyle factors, dietary intake is receiving the most attention and, as a result, comprehensive dietary-related interventions have been proclaimed to improve all aspects of MetS [7,8]. 

Despite the aforementioned evidence, to date, no specific diet has been shown to treat MetS in totality. Some studies have found that the Mediterranean [9,10] and DASH [11] diets can help to improve MetS and its components. However, the evidence suggests that a diet that is high in CHOs raises serum triglycerides, lowers serum high-density lipoprotein cholesterol (HDL-C) and impairs glucose metabolism, all of which are MetS components [12,13].

Through a Korean population-based survey among 6737 males and 8845 females, Kwon et al. (2018) [14] aimed to examine the role of CHOs and fat intake in MetS. The reported outcomes suggested that most males who consumed high proportions of CHOs in their diet presented with MetS. However, other studies produced conflicting findings between CHO intake and MetS risk [15,16,17]. For instance, in a cross-sectional study by Motamed et al. (2013) [18], which included 3800 men and women between the ages of 35 and 65 years, no significant relationship was indicated between CHO intake and MetS. Additionally, Eilat-Adar et al. examined the association between macronutrient intake, gender, MetS and insulin resistance (IR) among non-diabetic American Indians. In both men and women, no association was found between macronutrient intake and the prevalence of MetS [19]. A cross-sectional study of 1626 patients with a history of cardiovascular disease was conducted by Skilton et al. (2008) [20]. Conversely, their study showed that high CHO consumption reduced the prevalence of MetS.

Evidence has also shown that foods that are high in saturated fatty acids (SFAs) increase the risk of MetS [21]. Several studies have shown that the consumption of vegetable fat as opposed to animal fat is associated with a lower incidence of cardiometabolic disease and diabetes mellitus [22,23]. More importantly, there is evidence that the different types of fatty acids affect MetS differently. SFA consumption aggravates IR, whereas mono- and polyunsaturated fatty acid (MUFA and PUFA) consumption has the opposite effect [24].

It has been further indicated that high fat, salt and sugar, and low complex CHO eating habits among the South African population demonstrates a transition to a westernized diet over the last two decades [25,26,27]. This nutrition transition has been escalated by the dominance of big food (large retail markets) [28,29], which makes unhealthy food available in South African retail and informal markets. Moreover, the South African food environment, especially the food that is sold in the streets of South Africa, does not foster good health [30].

Despite the findings between macronutrients and the prevalence of MetS being contentious, we acknowledge that the proportions and types of macronutrients and fat in individuals’ diets may either impair health or result in the successful prevention and treatment of MetS and its components in South Africa—hence the current study. We ought to mention that a previous analysis on the same population examined MetS’s predictive power of anthropometric indices to determine the cut-off points and to identify male South African taxi drivers with MetS [4]

We, therefore, in the current study, conducted mathematical models to allow for the hypothetical substitution of one macronutrient or fatty acid for another, within an isoenergetic condition, to determine whether there is an association with the metabolic outcomes of male taxi drivers operating in the urban areas of South Africa. The relationship between macronutrient and fatty acid intake—with health markers such as waist circumference (WC), diastolic and systolic blood pressures (DBP and SBP), fasting blood glucose (FBG), dyslipidaemia (low-density lipoprotein (LDL-C)), HDL-C triglycerides and the clustering of MetS components—were included in the models. To the best of our knowledge, there is no study that has considered macronutrient substitution and the risk of MetS in South Africa. Furthermore, no literature could be found on a diet customized to prevent or reduce MetS within a South African context. Hence, we attempted to use the multivariable nutrient density substitution models to examine the association between macronutrient and fatty acid intake and MetS and its components in male mini-bus taxi drivers operating in the urban areas of South Africa. The outcomes of this research will be used to influence the interventions directed at preventing and reducing the prevalence of metabolic disorders in South Africa, especially among men working in the mini-bus taxi driving industry.

## 2. Materials and Methods

### 2.1. Study Population

This research formed part of a cross-sectional study, conducted with 237 male mini-bus taxi drivers, aged 20 years and older (mean age 39.9 ± 10.5 years), who operated in the Cape Town metropole, South Africa. For the current analysis, 185 taxi drivers who consumed street food at least three times a week; those who had at least one year of full-time employment as a mini-bus taxi driver; and those who donated a blood specimen, which was analyzed in the laboratory to estimate the MetS components, were included. Taxi drivers who had a history of underlying diseases including hypertension, kidney failure, hypo/hyperthyroidism liver diseases, known cardiovascular disease and diabetes mellitus were excluded from the analysis. More details of the study sample are presented elsewhere [4]. The protocol of the study was approved by the Biomedical Science Research Ethics Committee of the University of the Western Cape (ethics reference number: BM20/6/8). Informed consent was also obtained from all the participants.

### 2.2. Assessment of Socio-Economic and Lifestyle Variables

A previously validated questionnaire, which was used for the South African National Health and Examination Survey (SANHANES-1) [31], was used in this study to record information on sociodemographic characteristics—including age, sleeping duration, driving experience, money spent a day on SF, educational level, smoking status, race, marital status and alcohol intake. Age and driving experience were recorded in years. The sleeping duration was recorded in hours. Educational level was categorized as 1 = no schooling or primary, 2 = some high school and higher education. Smoking status was categorized as 1 = current smoker and 2 = non-smoker. Race was categorized as 1 = black and 2 = non-black. Marital status was categorized as 1 = single/separated or divorced and 2 = married or living as married. Two questions on alcohol consumption were also included. The first question was, ‘How often do you drink?’ The choices were as follows: (i) ‘I have never drunk alcohol’, (ii) ‘I no longer drink alcohol’; (iii) ‘I drink alcohol very rarely, less than once a week, 1 or 2 days a week, or 3 or 4 days a week’; (vii) ‘5 or 6 days a week’; and (viii) ‘every day’. A person was considered to be a current drinker for any choice between (iii) and (viii). The second question was, ‘On a day when you drink something with alcohol, how many standard drinks do you have/tend to have?’ The questionnaire defined a standard drink as ‘a small glass of wine, a regular beer can (330 mL), a shot of liquor or a cocktail.’ The choices were as follows: ‘13 or more standard drinks’; (ii) ‘9 to 12 standard drinks’; (iii) ‘7 to 8 standard drinks’; (iv) ‘5 to 6 standard drinks’; (v) ‘3 or 4 standard drinks’; and (vi) ‘1 or 2 standard drinks.’ The survey did not ask about the average duration of a drinking session.

The International Physical Activity Questionnaire (IPAQ) was used to measure the level of physical activity (PA). IPAQ included questions about the frequency and duration of vigorous-, moderate- and low-intensity physical activities, as well as how often and how long the taxi drivers had walked during the past week. For each category of walking— moderate and vigorous intensity—the physical activities were divided into the following four groups: work-related, transportation-related, household-related and leisure-related. Each type of activity (walking, moderate-intensity and vigorous-intensity) was counted separately by multiplying the number of days in a week by how long an average day is. The IPAQ core group [32] provided the following definitions of low, moderate and high levels of physical activity: low—no activity was reported or there was some activity but not enough to meet the criteria for the other activity categories; moderate—(a) 3 or more days of vigorous-intensity activity for at least 20 min per day, (b) 5 or more days of moderate-intensity activity or walking for at least 30 min per day, or (c) 5 or more days of any combination of walking, moderate-intensity or vigorous-intensity activities, which added up to at least 600 MET-minutes per week. Vigorous meant either (a) 3 or more days of vigorous-intensity activity, which added up to at least 1500 MET-minutes per week or (b) 7 days of walking, moderate-intensity activities or vigorous-intensity activities, which added up to at least 3000 MET-minutes per week. It has been reported that the IPAQ is reliable and valid [33].

### 2.3. Dietary Assessment

Dietary intake was assessed using 24-h dietary recalls. Dietary composition was analyzed using the South African Medical Research Council (SAMRC) FoodFinder 111 [34]. Analyzed diet components included macronutrients (total, grams and proportions of CHOs, protein and fat) and fatty acids (total and grams of SFAs, MUFAs and PUFAs). Other dietary intake variables were also assessed but not as dietary components of interest for the current analysis. 

### 2.4. Assessment of MetS Components

Bodyweight and height were measured using a platform scale and a fixed stadiometer, while wearing light clothing, without shoes. The body mass index (BMI) was calculated by dividing weight (kg) by height squared (m^2^). A non-elastic tape measure was used to measure the waist circumference (WC) at the narrowest point between the lower edge of the rib and the upper-iliac crest. Following a 10-min rest, systolic and diastolic blood pressure were taken three times on the right arm, using a standard sphygmomanometer. For analysis, the average of the 2 last blood pressure (BP) readings were used. 

To measure biochemical parameters, a venous fasting blood sample was sourced from the participants after an 8-h overnight fast and was kept on dry ice and transported to the laboratory for processing. The anthropometrical and biochemical measurements, and their categorization and analysis are explained in detail elsewhere [4].

### 2.5. Assessment of MetS

International Diabetes Federation (IDF) criteria was used to indicate MetS [35] among the participants. According to the IDF criteria, abdominal obesity (WC ≥94 cm for men), accompanied by two or more of the following cut-points are required to confirm MetS: triglycerides ≥1.7 mmol/L; systolic blood pressure (SBP) ≥130 mmHg, or diastolic blood pressure (DBP) ≥85 mmHg; fasting blood glucose (FBG) ≥5.6 mmol/L; and HDL-C <1.03 mmol/L, in men

### 2.6. Statistical Analysis

For continuous normally distributed data, descriptive statistics were reported as mean values and standard deviations (SD), and for continuous non-normally distributed data, as median and interquartile ranges (IQR). The independent samples *t*-test was used to compare participants with MetS and those without MetS. Multivariable nutrient density substitution models, as described by Skilton et al. (2008) [20] and Willett (1998) [36], were applied to investigate the relationship between macronutrient and fatty acid intakes and MetS. It should be noted that the substitution models represent an increase in the intake of one macronutrient, accompanied by a decrease in the intake of another macronutrient in isoenergetic conditions, in real life. For example, in the model used in the current research, a decrease in the percentage of calories consumed from total proteins were replaced by CHOs and fats, and vice versa. For fatty acids, the grams of MUFAs and PUFAs were replaced by SFAs, and vice versa. For the regression models, the presence of MetS and its components were entered as the dependent variables in the logistic regression models, while the linear regression model was used to analyze the continuous variables, including MetS and its components. To allow for the study of the effects of diet composition, all models were adjusted for total energy intake and alcohol intake. In each model, one macronutrient was included as a variable of interest and the respective macronutrient was excluded, while the other macronutrients were included as cofactors to adjust for their confounding effects in the models.

Results were interpreted as follows: if the outcome variable increased or decreased when the relevant macronutrient was removed from the model, then the relevant macronutrient was an isoenergetic substitute for the variable in question. Thus, for example, to examine the substitution of CHOs for protein, protein was included as an independent variable of choice/interest; CHOs were excluded from the model; and the model was adjusted for fat and energy intake. In this case, the odds ratio (OR) for protein represented substituting protein for an isoenergetic quantity of dietary protein. All macronutrients were entered into the models as percentages (%E). Furthermore, all the models were first adjusted for lifestyle factors, e.g., age, race, marital status, level of education, smoking status, driving experience, alcohol intake and physical activity. BMI was used to adjust for models that examined the non-abdominal obesity of the MetS, including elevated triglycerides, low HDL-C, as well as elevated BP and FBG. A two-tailed *p* value of <0.05 was deemed statistically significant. IBM SPSS Statistics for Windows, version 28.0, was used to analyze all data (IBM Corp).

## 3. Results

### Descriptive Analysis

The sociodemographic and physical characteristics of the study participants are shown in Table 1. The mean age of the 185 men was 39.9 ± 10.7 years. These men had 9 years of driving experience as taxi drivers, slept an average of 6 h per night and displayed mean WC, WhtR, BMI and FBG levels above the normal ranges. Participants with MetS were significantly (*p* < 0.001) older (43.7 ± 10.3 vs. 37.3 ± 10.2 years) and had more driving experience (11.7 ± 8.4 vs. 7.2 ± 6.1 years), compared to those without MetS. Moreover, participants with MetS had significantly (*p* < 0.001) higher mean values for WC (110.8 ± 16.7 vs. 90.7 ± 14.5 cm); FBG (7.9 ± 4.8 vs. 5.3 ± 1.1); SBP (141.5 ± 18.8 vs. 127.4 ± 13.3); DBP (92.7 ± 13.9 vs. 79.1 ± 9.1); triglycerides (1.9 ± 1.5 vs. 1.0 ± 0.4); BMI (32.7 ± 5.9 vs. 25.7 ± 5.2); WHtR (0.6 ± 0.1 vs. 0.5 ± 0,1); and low HDL-C (1.0 ± 0.3 vs. 1.2 ± 0.4), compared to those without MetS.

Table 2 shows the participants’ ranges of macronutrients and other nutrient intakes by the MetS status. The overall median energy intake was 11,059.0 kJ/d (IQR 7441.0–17,195.5). The overall median intakes of CHOs, protein and fats—expressed as percentage of energy (%E)—were 57.3 (IQR 49.6–67.7), 14.9 (IQR 12.2–17.8) and 25.0 (IQR 17.0–33.4). No significant associations were observed between nutrient intake and the prevalence of MetS. Despite no significant associations being found, the median values of the variables among participants with MetS were higher than those without MetS. Finally, the median values for plant protein, CHOs (% E), protein (% E) and PUFAs were similar between the participants with and those without MetS.

No significant associations (*p* > 0.05) were observed between the %E derived from macronutrient intake and the prevalence of MetS, abdominal obesity (WC), raised triglyceride or FBG (Table 3)—with the exception of the energy derived from protein, which was significantly associated with a raised BP adjusted odds ratio (AOR) (AOR 1.108, 95%CI 1.026–1.197, *p* = 0.007) and low HDL-C (AOR 0.914, 95%CI 0.844–0.988, *p* = 0.025). In this case, it was shown that an increase in protein consumption significantly (*p* < 0.007) increased the likelihood of elevated BP by almost 11% and significantly lowered the likelihood of reduced HDL cholesterol by 8.6%. However, the association of SFA consumption and FBG only tended to significance (AOR 1.096, 95%CI 1.099–1.203, *p* = 0.053).

Table 4 presents the outcomes of the substitution models for protein, CHOs and the total fat. The model was adjusted for age, race, physical activity, marital status, level of education, driving experience, smoking status, total energy, alcohol intake and BMI. Substituting CHOs and total fat for protein, decreased the likelihood of reduced HDL-C by 7% and 7.2% (AOR 0.930 95%CI 0.875–0.989, *p* = 0.021 and AOR 0.928 95%CI 0.874 0.985, *p* = 0.014), respectively. Moreover, substituting total fat for CHOs reduced the likelihood of elevated BP by 2.6% (AOR 0.974 95%CI 0.956–0.992, *p* = 0.005) when dietary CHOs were substituted with protein. 

Table 5 presents the outcomes of the substitution models for protein, CHOs and total fat. When no other lifestyle factors were considered, with the exception of keeping the total energy constant (i.e., adjusted for total food energy and alcohol intake), the intake of protein instead of CHOs and total fat, reduced the likelihood of raised triglycerides by 6.7% and 6.6% (AOR 0.933 95%CI 0.880–0.990, *p* = 0.021 and AOR 0.934 95%CI 0.892–0.978, *p* = 0004), respectively. Moreover, the intake of CHOs instead of protein and total fat reduced the likelihood of elevated BP by 2.2% and 2.8% (AOR 0.978 95%CI 0.986–0.991, *p* = 0.001 and AOR 0.972 95%CI 0.957–0.988, *p* = 0.001) respectively. The replacement of protein by total fat only tended to significance (*p* = 0.050), while demonstrating an increased likelihood of raised triglycerides (OR 1.974 95%CI 0.94–0.999).

In Table 6, we repeated the multivariable nutrient density substitution models, however, in this case, using the outcomes for fatty acids. The model was adjusted for age, race, physical activity, marital status, level of education, driving experience, smoking status, and total energy and alcohol intake. No significant associations were observed between the fatty acid outcomes with MetS and its components when MUFAs and PUFAs were substituted for SFAs, and when PUFAs were substituted for MUFAs. However, substituting SFAs for PUFAs significantly decreased the likelihood of elevated BP by 7% (AOR 0.930 95%CI 0.858 0.989, *p* = 0.047). The outcomes for abdominal obesity, raised triglyceride, reduced HDL-C, elevated BP and FBG remained unchanged when SFAs were substituted with MUFAs and MUFAs was substituted with PUFAs.

In Table 7, the multivariable nutrient density substitution models are reported. The model was adjusted for total energy and alcohol intake. Substituting MUFAs for SFAs significantly increased the likelihood of elevated BP by 9.8% (AOR 1.098 95%CI 0.831–0.992, *p* = 0.033), while substituting SFAs for PUFAs decreased the likelihood of elevated BP by 9.5% (AOR 0.905 95%CI 0.845 0.969, *p* = 0.004).

%E—percentage energy; MetS—metabolic syndrome; WC—waist circumference; HDL-C—high-density lipoprotein cholesterol; FBG—fasting blood glucose; BP, blood pressure; SFA—saturated fatty acid; MUFA—monounsaturated fatty acid; PUFA—polyunsaturated fatty acid. MUFA, PUFA and SFA intakes were included as either the variable of interest (↑); were excluded from the model, when the fatty acid was substituted for (↓); or were adjusted for as a covariate. The MetS was defined using the IDF definition. Fatty acids were entered as percentage of total energy intake. The results are presented as OR for the presence of the MetS per change in the proportion of dietary energy. All models were adjusted for age, race, physical activity, marital status, level of education, driving experience, smoking status, total energy, alcohol intake, CHO intake, protein intake and *trans* fatty acid intake. † Also adjusted for BMI.

Finally, Table 8 presents the outcomes of linear regression models to show the association between macronutrient and fatty acid consumption substitution and MetS and its components. In this case, substituting protein for total fat was significantly associated with elevated FBG (β 0.491 95%CI 0.009, 0.972, *p* = 0.046). Moreover, when MUFAs were substituted with SFAs, a higher FBG level was observed (β 1.105 95%CI −2.185, −0.025, *p* = 0.045). This meant that a diet that was high in total fat and saturated fat tended to increase FBG levels.

## 4. Discussion

In the present study, we conducted real life simulations of macronutrient and fatty acid substitution, using the multivariable nutrient density substitution models. Taxi drivers who presented with MetS were older, had more driving experience, and presented with larger body sizes and body fat centralization. They also had higher levels of FBG, SBP and triglycerides, but lower levels of HDL-C compared to their counterparts without MetS. These outcomes are corroborated by substantiated international evidence [37], where it has been highlighted that the likelihood of MetS and other metabolic disorders is higher in individuals who are employed in the taxi driving industry [37], those who are older [38], and those who present with larger body sizes and body fat centralization [39,40].

While no significant associations were observed between nutrient intake and the prevalence of MetS, the overall median energy intake of the participants was above the recommended dietary reference intakes for average men (i.e., men with the average BMI of 22.5 m^2^/kg, which is equal to 10,626 kJ), based on the Food and Nutrition Board (2004) [41]. Kolahdooz et al. (2013) [42] and Wentzel-Viljoen and Kruger (2010) [43] found similar outcomes for total energy intakes that were above the normal range (i.e., 11,159 kJ and 15,485 kJ among urban South African men, respectively). The elevated total energy intake among the taxi drivers is a cause for concern, as evidence indicates that a higher energy intake than energy output increases the risk of excessive weight gain, especially abdominal fat accumulation [44]. Taxi drivers were sedentary, based on the outcomes of their physical activity/inactivity, where 75.7% engaged in low physical activity, 60.5% slept less than 6 h per night and, on average, sat for 3 h daily in their taxis, without driving. Corroborating other South African evidence [42,45], our outcomes suggested that most of the total energy consumed by urban black South African men come from CHOs (65%), while the energy derived from fat (20%) and protein (10%) were at the lower extreme ends of the recommended macronutrient energy intake for men, based on the recommendations of the Food and Nutrition Board (2004) [41].

The results of dietary macronutrient composition—especially the amount and quality of CHOs, protein, fats, and their impacts on health—have received more attention in recent years. Over the past two decades there has been growing evidence that suggests that weight loss diets that restrict CHOs are more effective, prevent metabolic disorders [46]. In fact, Volek et al. (2008) [12] and Jung and Choi (2017) [13] argued that diets that are high in CHOs raise serum triglycerides, lower serum HDL-C and impair glucose metabolism. Furthermore, Kwon et al. (2018) [14] suggested that males who consume high proportions of CHOs in their diet present with MetS. As in similar studies by Eilat-Adar et al. (2008) [19] and Motamed et al. (2013) [18], our current research did not show such significant associations. We also have to highlight the fact that the increased weight reduction benefits of restricting dietary CHOs, as opposed to restricting dietary fat, as highlighted by the popular and growing body of evidence [47,48,49,50], has been challenged by the review article of Naude et al., (2014) [46]. Here, the authors argued that there is little or no difference in metabolic disorder prevalence among overweight and obese adults who follow low CHO diets. Moreover, in their response to the questions raised about their review, Naude et al. (2014) [46] further clarified that a result of 780 g more weight loss after a 3-to-6-month intervention of a low CHO diet compared to other weight loss diets cannot be practically and clinically concluded as more effective in weight loss than the comparative diet. However, it is also important to mention that, in the current analysis, we observed that substituting protein and fat for CHOs, reduced the likelihood of elevated BP. These findings are congruent to those of Teunissen-Beekman et al. (2013) [51], where it was shown that BP decreases more after a high-CHO meal than after a high-protein meal, especially among overweight adults with elevated BP. Indeed, the majority of taxi drivers were overweight and obese, and their mean BMI and BP were above the normal ranges. Moreover, Savoia et al. (2021) [44] argued that high potassium, antioxidants and fiber-rich sources of CHOs (i.e., fruits, vegetables, potato starch and grains) may be the compounds that contribute to the reduction of blood pressure.

In the current analysis we also showed that the energy intake derived from protein raised BP. However, similar studies presented contrasting evidence. They suggested that an increase in dietary protein leads to a decrease in mean SBP and DBP by 4.9 and 2.7 mmHg, respectively [51]. A study by Rebholz et al. (2012) [52] also showed a significant reduction in BP (i.e., 1.8 mmHg for SBP and 1.2 mmHg for DBP decrease) when dietary protein was consumed, compared to CHO or fat consumption. He et al. (2011) [53] also demonstrated a significant reduction in SBP (2.0 mm Hg) for both soy protein and milk protein, in comparison with CHOs. According to Melson et al. (2019) [54] and Astrup et al. (2015) [47], fat, as well as protein, increases satiety, which leads to a reduced energy intake with concomitant weight reduction and, subsequently, to lower BP. In addition, protein is thought to increase postprandial energy expenditure due to the higher metabolic processes needed to metabolize fat and CHOs.

The consumption of protein by the taxi drivers participating in our research was shown to be beneficial, as it significantly lowered the likelihood of reduced HDL-C. These outcomes were corroborated by the findings of Pasiakos et al. (2015) [55]; Dong et al. (2013) [56]; Layman et al. (2009) [57]; and the 2010 Dietary Guidelines for Americans [58], where it is reported that the cardiometabolic benefits of high protein diets are often greater than those observed when consuming low-fat and high-carbohydrate diets. However, it should be noted that most of the studies in the literature that have reported the benefits of a high protein intake were conducted on overweight and obese adults, who were undergoing well-controlled weight loss interventions [59]. The evidence suggests that an increased protein intake at the expense of CHOs is generally considered to reduce cardiometabolic disorders, through glycemic [56] and blood lipid [57] regulation. This was observed especially among adults with a high cardiovascular disease risk following a controlled high-protein, low-carbohydrate weight loss diet. Additionally, habitually consuming a high-protein diet was associated with higher HDL-C (and lower adiposity) levels, regardless of total dietary energy, CHO and fat intake. Pasiakos et al. (2014) [55] also argued that the intrinsic properties of protein, unrelated to its energy content, appear to be partially responsible for these effects. Moreover, Hooseini-Esfahan et al. (2019) [60] emphasized that a higher proportion of dietary protein, especially plant-derived protein, in place of CHOs and fat is beneficial to weight loss and reduces body fat centralization. As we have shown in our models—in which we adjusted for socio-economic status, lifestyle and other physiological confounders—there is a robust association between dietary protein and HDL-C, which is particularly intriguing. However, the mechanism by which dietary protein is associated with the upregulation of HDL-C production still requires further investigation.

In the current study, we used the multivariable nutrient density substitution model and we showed that the substitution of dietary protein for energy derived from fat resulted in a significant upregulation of triglycerides. We also showed that a diet that was high in total fat increased the FBG in the taxi drivers. In contrast to our findings, Skilton et al. (2008) [20] reported lower odds of elevated WC when they replaced CHOs with fats. They further reported that an isoenergetic increase in fat intake at the expense of protein reduced the odds for MetS. Considering this contrasting evidence to our outcomes, similar trials to confirm recent findings need to be conducted.

Our findings on BP and PUFAs are similar to those of Jovanovski et al., (2014) [61] and Guo et al., (2014) [62], who argued that PUFA intake lowers BP and MetS risk. Furthermore, SFAs and unsaturated fatty acids are well known for their role in the risk and prevention of MetS, respectively [63]. According to Imamura et al. (2016) [64], PUFAs, as well as MUFAs, reduce IR and also lower LDL-C and apolipoproteins. Guerendiain et al. (2018) [65] further argued that trans FAs and SFAs appear to increase IR and glucose intolerance, which are the major contributors to MetS. Similar significant relationships between MUFA intake and DBP were reported [66], where the authors reported that MUFA intake—especially oleic acid from vegetable sources—may contribute to the prevention and control of adverse BP levels in general populations. More evidence [67,68,69] indicated, that PUFA and MUFA intakes reduce MetS components such as triglycerides, HDL-C, glucose and BP levels.

Considering the foregoing, it is important to note that the contradictions we found between the outcomes of the current study and the outcomes of the majority of the literature presented, could be ascribed to the fact that the literature outlined above was based on real-life interventions, whereas ours used multivariable nutrient density substitution models. Moreover, none of the evidence we provided attempted multivariable nutrient density substitution models using taxi driver outcomes, or outcomes for the people involved in the taxi driving industry. However, it is important to note that these populations are at risk of cardiovascular diseases since they are exposed to unhealthy food [30,70]. Moreover, their lifestyles—including physical inactivity, excessive alcohol intake and cigarette smoking [71]—put them at an increased risk of MetS. Hence, further research of this kind is needed to identify whether macronutrients and fatty acids should be consumed to reduce MetS and its components, especially in real life.

### Limitations

While the aforementioned study has managed to highlight several strengths, there were a number of limitations to be considered when interpreting our results. Firstly, because we used a cross-sectional study design, we could not make definite conclusions about the associations we found between macronutrient and fatty acid intake and MetS risk. As such, we recommend further prospective studies in real life, which will reveal the actual associations between macronutrient and fatty acid consumption and MetS and its components. Secondly, it would be inappropriate to generalize the study outcomes to the entire urban South African population, given that the data were only based on mini-bus taxi drivers, who consumed street food at least three days a week. Thirdly, while the use of a substitution model is justified when studying the health effects of different macronutrients in isoenergetic conditions, it must be acknowledged that this approach is only a mathematical model for dietary intake and not a real-life situation.

## 5. Conclusions

The present study indicated that South African, male mini-bus taxi drivers, who operate in the urban areas, consume total dietary energy that is above the recommended dietary allowances and are at risk of MetS. In an isoenergetic state, the taxi drivers’ diets— which were high in protein, CHOs and PUFAs—reduced triglycerides and BP, respectively. Whereas, when their diets were high in total fat and SFAs the opposite effects were observed, with an added disadvantage of elevated FBG. It is, however, important to note that these outcomes were produced using mathematical models. As such, we recommend further prospective studies in real life, which will reveal the actual associations between macronutrient and fatty acid consumption and MetS and its components.

## Figures and Tables

**Table 1 ijerph-19-15452-t001:** Characteristics of the study participants in relation to their MetS status.

			IDF MetS				
	Entire Cohort (*n* = 185)		No (*n* = 108)		Yes (*n* = 77)		
	Mean	SD	Mean	SD	Mean	SD	*p*-Values
Age (years)	39.9	10.7	37.3	10.2	43.7	10.3	<0.001
Years in the taxi driving industry (*n*)	9.1	7.4	7.2	6.1	11.7	8.4	<0.001
Sleep duration (hours)	6.1	1.1	6.1	1.0	6.2	1.2	0.624
WC_(cm)	99.1	18.3	90.7	14.5	110.8	16.7	<0.001
Weight_(kg)	84.6	20.4	75.5	16.4	97.3	18.7	<0.001
Height_(cm)	171.9	8.1	171.4	7.2	172.6	9.3	0.346
FBG_(mmol/L)	6.4	3.5	5.3	1.1	7.9	4.8	<0.001
SBP_(mmHg)	133.3	17.2	127.4	13.3	141.5	18.8	<0.001
DBP_(mmHg)	84.7	13.2	79.1	9.1	92.7	13.9	<0.001
hsCRP_(mg/L)	4.9	8.4	4.3	9.6	5.6	6.5	0.287
LDL-C _(mmol/L)	2.8	0.82	2.7	0.8	2.8	0.8	0.336
Triglycerides_(mmol/L)	1.3	1.1	1.0	0.4	1.9	1.5	<0.001
HDL-C_(mmol/L)	1.1	0.3	1.2	0.4	1.0	0.3	<0.001
BMI_(kg/m^2^)	28.6	6.5	25.7	5.2	32.7	5.9	<0.001
WHtR	0.6	0.1	0.5	0.1	0.6	0.1	<0.001

SD—standard deviation; IDF—International Diabetes Federation; WC—waist circumference; FBG—fasting blood glucose; SBP—systolic blood pressure; DBP—diastolic blood pressure; hsCRP—high-sensitivity C-reactive protein; LDL-C—low-density lipoprotein cholesterol; HDL-C—high-density lipoprotein; BMI—body mass index; WHtR—waist-to-height ratio.

**Table 2 ijerph-19-15452-t002:** Median and interquartile ranges of macronutrients and other nutrient intakes by the MetS status.

			IDF MetS Classification				
	Entire Cohort (*n* = 185)		No (*n* = 108)		Yes (*n* = 77)		
	Median	IQR	Median	IQR	Median	IQR	*p*-Values
Moisture (g)	1134.3	838.5–1621.1	1096.9	831.7–1375.7	1270.2	822.4–1805.8	0.257
Energy (kJ/d)	11,059.0	7441.0–17,195.5	10,900.0	7852.3–15,319.5	12,340.0	5861.0–18,765.5	0.704
Total protein (g)	96.7	67.0–143.3	95.7	75.4–127.6	115.2	57.2–180.5	0.360
Animal protein (g)	46.0	25.6–78.0	45.9	27.5–65.4	53.3	22.3–90.0	0.220
Plant protein (g)	27.9	17.3–42.4	28.5	18.5–41.2	28.8	16.3–46.0	0.515
Available CHOs (g)	349.8	225.6–522.7	349.8	237.7–459.5	375.7	193.8–579.1	0.958
Total CHOs (g/d)	379.6	252.3–558.1	378.2	259.7–497.2	409.9	206.5–651.2	0.957
CHO (% E)	57.3	49.6–67.7	57.9	50.0–68.3	57.6	48.8–66.0	0.339
Protein (% E)	14.9	12.2–17.8	14.9	11.5–17.9	15.0	13.1–17.7	0.668
MUFA (g)	23.2	13.1–46.6	21.8	14.1–42.8	27.0	9.5–53.9	0.449
PUFA (g)	20.4	9.2–41.2	20.9	9.5–37.9	21.4	7.1–45.4	0.344
Total fat (g)	83.0	40.8–127.8	70.2	43.1–120.3	87.8	29.8–139.5	0.373
Fat (% E)	25.0	17.0–33.4	24.3	17.0–33.5	27.6	17.3–33.6	0.661
SFA	18.8	10.6–30.3	18.4	11.7–27.3	21.3	9.7–32.9	0.817
SFA (% E)	6.4	4.6–8.4	6.3	4.6–8.2	6.6	4.5–8.5	0.988
MUFA (% E)	8.1	5.5–11.6	7.7	5.5–11.3	8.8	5.4–11.6	0.189
PUFA (% E)	6.7	4.3–10.4	6.7	4.3–10.2	6.7	3.7–11.0	0.983
Starch (g)	8.2	0.0–16.7	10.1	3.9–16.1	7.8	0.0–16.9	0.741
Added sugar	0.0	0.0–0.3	0.0	0.0–0.3	0.0	0.0–0.4	0.406
Total sugar	35.7	12.4–68.7	36.4	13.0–66.0	38.5	8.9–71.6	0.698
Total trans FA	0.4	0.1–1.0	0.4	0.1–0.9	0.6	0.1–1.1	0.757
Total fiber	33.0	22.0–50.3	31.5	24.2–45.5	36.2	18.4–54.9	0.595
Insoluble fiber	2.3	1.1–4.2	2.5	1.4–4.2	1.9	0.3–4.2	0.480
Soluble fiber	1.7	0.9–3.1	1.8	1.1–2.9	1.4	0.3–3.2	0.607

IQR—interquartile range (i.e., 25th–75th percentile); MetS—metabolic syndrome; IDF—International Diabetes Federation; %E—percentage energy; PUFA—polyunsaturated fatty acid; MUFA—monounsaturated fatty acid; CHO—carbohydrate; SFA—saturated fatty acid.

**Table 3 ijerph-19-15452-t003:** The association between dietary CHOs, protein and fat and MetS and its components.

	MetS			Abnormal WC			Raised Triglycerides †			Low HDL-C †			Raised BP †			Elevated FBG †		
	AOR	95%CI	*p*	AOR	95%CI	*p*	AOR	95%CI	*p*	AOR	95%CI	*p*	AOR	95%CI	*p*	AOR	95%CI	*p*
CHO_% E	1.001	0.998–1.004	0.650	1.008	0.984–1.032	0.517	1.001	0.995–1.004	0.612	1.011	0.985–1.038	0.403	0.994	0.970–1.019	0.639	1.000	0.998–1.002	0.738
Protein _% E	1.016	0.945–1.093	0.667	0.991	0.927–1.060	0.991	1.035	0.954–1.124	0.406	0.914	0.844–0.988	0.025	1.108	1.026–1.197	0.007	1.001	0.937–1.069	0.982
FAT_% E	1.005	0.973–1.038	0.763	0.991	0.964–1.020	0.542	1.027	0.989–1.066	0.168	0.996	0.966–1.027	0.792	1.000	0.972–1.030	0.977	1.024	0.995–1.053	0.106
SFA_% E	1.019	0.925–1.123	0.699	0.965	0.883–1.055	0.434	1.113	0.990–1.251	0.073	0.995	0.906–1.093	0.913	1.012	0.925–1.105	0.795	1.096	1.099–1.203	0.053
MUFA_% E	1.026	0.969–1.086	0.374	1.007	0.990–1.024	0.410	1.002	0.994–1.010	0.679	1.002	0.991–1.013	0.737	1.001	0.993–1.009	0.856	1.063	0.994–1.136	0.074
PUFA_% E	0.977	0.910–1.048	0.515	1.014	0.951–1.081	0.670	1.001	0.922–1.086	0.986	1.001	0.935–1.072	0.974	0.965	0.903–1.033	0.304	0.997	0.938–1.060	0.928

SFA—saturated fatty acid; HDL-C—high-density lipoprotein cholesterol; WC—waist circumference; BP—blood pressure; CHO—carbohydrate; MUFAs— monounsaturated fatty acids; PUFAs—polyunsaturated fatty acids. CHO, protein, fat, SFA, MUFA and PUFA were included as independent variables of interest. The MetS were defined using the IDF definition. Macronutrients and fatty acids were entered as a percentage of total energy intake. The results are presented as OR for the presence of the MetS, per change in the proportion of dietary energy. All models were adjusted for age, race, physical activity, marital status, level of education, smoking status, driving experience, total energy and alcohol intake. OR—odds ratios, 95%; CI—95% confidence intervals. † Also adjusted for BMI.

**Table 4 ijerph-19-15452-t004:** The association between macronutrient (CHOs, protein and total fat) dietary intake and the MetS and its components. The model was adjusted for all lifestyle factors and BMI.

	MetS			Abnormal WC			Raised Triglycerides †			Low HDL-C †			Raised BP †			Elevated FBG †		
	AOR	95%CI	*p*	AOR	95%CI	*p*	AOR	95%CI	*p*	AOR	95%CI	*p*	AOR	95%CI	*p*	AOR	95%CI	*p*
↑CHO_%E ↓Protein_%E	1.001	0.997–1.005	0.681	1.004	0.963–1.047	0.842	1.002	0.957–1.050	0.926	1.021	0.981–1.062	0.311	0.988	0.951–1.027	0.552	1.001	0.997–1.004	0.777
↑CHO_%E ↓FAT_%E	1.001	0.997–1.005	0.666	1.008	0.983–1.035	0.532	1.001	0.997–1.005	0.639	1.001	0.995–1.006	0.833	0.974	0.956–0.992	0.005	1.000	0.998–1.002	0.737
↑FAT_%E ↓Protein_%E	1.005	0.973–1.038	0.767	1.001	0.954–1.050	0.974	1.027	0.972–1.085	0.338	1.008	0.968–1.064	0.746	1.002	0.959–1.048	0.696	1.020	0.995–1.053	0.151
↑Protein_%E ↓CHO_%E	0.995	0.921–1.075	0.901	0.973	0.914–1.035	0.381	0.940	0.874–1.011	0.059	0.930	0.875–0.989	0.021	1.012	0.951–1.077	0.700	0.975	0.919–1.035	0.405
↑FAT_%E ↓CHO_%E	1.003	0.969–1.038	0.875	0.987	0.955–1.021	0.450	1.002	0.964–1.041	0.926	0.997	0.965–1.029	0.834	0.976	0.943–1.011	0.179	1.022	0.989–1.055	0.190
↑Protein_%E ↓FAT_%E	1.002	0.930–1.080	0.958	0.965	0.912–1.021	0.212	0.947	0.885–1.014	0.118	0.928	0.874–0.985	0.014	1.026	0.965–1.090	0.412	0.994	0.942–1.048	0.823

%E—percentage energy; CHO—carbohydrate; MetS—metabolic syndrome; WC—waist circumference; BP—blood pressure; FBG—fasting blood glucose; HDL-C—high-density lipoprotein cholesterol. Substitution model, where CHOs, protein and fat were included as the variable of interest (↑), or were excluded from the model, when the macronutrient was substituted for (↓), or was adjusted for as a covariate. The MetS was defined using the IDF definition. Macronutrients were entered as a percentage of total energy intake. The results were presented as OR for the presence of MetS per change in the proportion of dietary energy. The OR for the ‘opposite’ substitution was the inverse of that presented, and the *p* value was the same. For example, the OR for the MetS when substituting fat for CHOs = (OR 1.005 95% CI 0.963–1.054, *p* = 0.681). All models were adjusted for age, race, physical activity, marital status, level of education, driving experience, smoking status, and total energy and alcohol intake. ORs—odds ratios, 95%; CIs—95% confidence intervals. † Also adjusted for BMI.

**Table 5 ijerph-19-15452-t005:** The association between macronutrient (CHOs, protein and total fat) intake and the MetS and its components. The model was adjusted for total energy and alcohol intake.

	MetS			Abnormal WC			Raised Triglycerides			Low HDL-C			Raised BP			Elevated FBG		
	AOR	95%CI	*p*	AOR	95%CI	*p*	AOR	95%CI	*p*	AOR	95%CI	*p*	AOR	95%CI	*p*	AOR	95%CI	*p*
↑CHO_%E ↓Protein_%E	1.001	0.999–1.002	0.441	1.001	0.997–1.004	0.662	1.001	0.999–1.002	0.284	1.007	0.996–1.018	0.191	0.978	0.966–0.991	0.001	1.001	0.999–1.002	0.519
↑CHO_%E ↓FAT_%E	1.001	0.999–1.002	1.002	1.001	0.998–1.003	0.616	1.001	0.999–1.002	0.387	1.012	0.998–1.027	0.088	0.972	0.957–0.988	0.001	1.000	0.999–1.002	0.507
↑FAT_%E ↓Protein_%E	0.983	0.961–1.005	0.131	0.990	0.968–1.012	0.379	1.974	0.948–0.999	0.050	1.002	0.979–1.025	0.870	0.977	0.952–1.002	0.066	1.004	0.982–1.026	0.738
↑Protein_%E ↓CHO_%E	0.975	0.930–1.023	0.302	1.012	0.965–1.062	0.611	0.933	0.880–0.990	0.021	0.992	0.947–1.040	0.742	0.984	0.937–1.035	0.537	0.989	0.944–1.036	0.630
↑FAT_%E ↓CHO_%E	0.994	0.966–1.023	0.677	0.986	0.958–1.015	0.352	1.002	0.969–1.036	0.924	1.011	0.982–1.040	0.464	0.968	0.938–0.999	0.043	1.009	0.981–1.038	0.527
↑Protein_%E ↓FAT_%E	0.968	0.933–1.004	0.083	0.997	0.961–1.034	0.856	0.934	0.892–0.978	0.004	0.976	0.928–1.025	0.325	1.013	0.963–1.066	0.608	0.998	0.963–1.034	0.899

%E—percentage energy; CHO—carbohydrate; MetS–metabolic syndrome; WC–waist circumference; BP—blood pressure; FBG—fasting blood glucose; HDL-C—high-density lipoprotein cholesterol. Substitution model where CHOs, protein and fat were included as the variable of interest (↑) or were excluded from the model, when the macronutrient substituted for (↓) or were adjusted for as a covariate. The MetS was defined using the IDF definition. Macronutrients were entered as a percentage of total energy intake. The results are presented as AOR for the presence of the MetS per change in the proportion of dietary energy. The AOR for the ‘opposite’ substitution was the inverse of that presented, and the *p* value was the same. For example, the AOR for the MetS when substituting fat for CHOs = (AOR 1.005 95% CI 0.963–1.054, *p* = 0.681). All models were adjusted for total food energy and alcohol intake. AOR—adjusted odds ratios, 95%; CIs—95% confidence intervals.

**Table 6 ijerph-19-15452-t006:** The association between dietary fatty acids and the MetS and its components. The model was adjusted for all lifestyle factors and BMI.

	MetS			Abnormal WC			Raised Triglycerides †			Low HDL-C †			Raised BP †			Elevated FBG †		
	AOR	95%CI	*p*	AOR	95%CI	*p*	AOR	95%CI	*p*	AOR	95%CI	*p*	AOR	95%CI	*p*	AOR	95%CI	*p*
↑SFA_%E ↓MUFA_%E	0.830	0.655–1.053	0.124	0.999	0.817–1.223	0.996	1.060	0.825–1.363	0.649	1.127	0.891–1.427	0.319	0.828	0.662–1.034	0.096	1.028	0.837–1.127	0.791
↑SFA_%E ↓PUFA_%E	0.985	0.837–1.158	0.852	0.945	0.818–1.092	0.443	1.108	0.926–1.326	0.262	1.093	0.942–1.292	0.299	0.943	0.805–1.104	0.463	1.0102	0.943–1.286	0.221
↑MUFA_%E ↓PUFA_%E	1.022	0.961–1.088	0.481	1.010	0.982–1.038	0.498	1.001	0.993–1.009	0.821	1.003	0.992–1.013	0.632	1.000	0.992–1.008	0.993	1.061	0.917–1.227	0.429
↑MUFA_%E ↓SFA_%E	1.009	0.982–1.037	0.519	1.004	0.991–1.018	0.546	0.997	0.988–1.005	0.436	1.002	0.993–1.011	0.649	0.997	0.988–1.006	0.503	1.036	0.961–1.117	0.356
↑PUFA_%E ↓SFA_%E	0.950	0.882–1.023	0.174	1.007	0.937–1.083	0.841	0.945	0.865–1.031	0.202	0.981	0.917–1.050	0.587	0.930	0.858–0.989	0.047	0.978	0.907–1.055	0.565
↑PUFA_%E ↓MUFA_%E	0.960	0.892–1.034	0.286	1.022	0.948–1.102	0.569	0.944	0.864–1.032	0.205	0.987	0.921–1.058	0.715	0.938	0.864–1.019	0.132	0.987	0.920–1.058	0.712

%E—percentage energy; MetS—metabolic syndrome; WC—waist circumference; HDL-C—high-density lipoprotein cholesterol; FBG—fasting blood glucose; BP, blood pressure; SFA—saturated fatty acid; MUFA—monounsaturated fatty acid; PUFA—polyunsaturated fatty acid. MUFA, PUFA and SFA intakes were included as either the variable of interest (↑); were excluded from the model, when the fatty acid was substituted for (↓); or were adjusted for as a covariate. The MetS was defined using the IDF definition. Fatty acids were entered as percentage of total energy intake. The results are presented as OR for the presence of the MetS per change in the proportion of dietary energy. All models were adjusted for age, race, physical activity, marital status, level of education, driving experience, smoking status, total energy, alcohol intake, CHO intake, protein intake and *trans* fatty acid intake. † Also adjusted for BMI.

**Table 7 ijerph-19-15452-t007:** The association between dietary fatty acids and MetS and its components. The model was adjusted for total food energy and alcohol intake.

	MetS			Abnormal WC			Raised Triglycerides			Low HDL-C			Raised BP			Elevated FBG		
	OR	95%CI	*p*	OR	95%CI	*p*	OR	95%CI	*p*	OR	95%CI	*p*	OR	95%CI	*p*	OR	95%CI	*p*
↑SFA_%E ↓MUFA_%E	0.956	0.886–1.032	0.252	0.936	0.867–1.011	0.094	0.971	0.891–1.057	0.492	1.015	0.944–1.091	0.694	1.098	0.831–0.992	0.033	1.018	0.946–1.094	0.639
↑SFA_%E ↓PUFA_%E	0.922	0.844–1.006	0.069	0.936	0.867–1.011	0.092	0.944	0.868–1.028	0.188	1.020	0.952–1.092	0.576	0.874	0.801–0.953	0.202	0.984	0.893–1.085	0.751
↑MUFA_%E ↓PUFA_%E	1.018	0.973–1.065	0.442	1.011	0.982–1.040	0.457	0.999	0.991–1.007	0.834	1.002	0.994–1.010	0.649	1.000	0.992–1.008	0.996	1.024	0.952–1.100	0.528
↑MUFA_%E ↓SFA_%E	1.010	0.988–1.031	0.383	1.005	0.992–1.018	0.438	0.999	0.991–1.007	0.751	1.002	0.994–1.010	0.613	0.998	0.990–1.007	0.656	1.023	0.968–1.081	0.423
↑PUFA_%E ↓SFA_%E	0.957	0.901–1.016	0.149	1.008	0.952–1.069	0.776	0.930	0.862–1.003	0.060	1.022	0.966–1.082	0.448	0.905	0.845–0.969	0.004	0.981	0.921–1.046	0.561
↑PUFA_%E ↓MUFA_%E	0.978	0.917–1.042	0.485	1.035	0.971–1.103	0.293	0.939	0.866–1.018	0.127	1.019	0.958–1.084	0.548	0.934	0.868–1.006	0.072	0.989	0.930–1.051	0.717

%E—percentage energy; MetS—metabolic syndrome; WC—waist circumference; HDL-C—high-density lipoprotein cholesterol; FBG—fasting blood glucose; BP, blood pressure; SFA—saturated fatty acid; MUFA—monounsaturated fatty acid; PUFA—polyunsaturated fatty acid. MUFA, PUFA and SFA intake were included as either the variable of interest (↑); were excluded from the model, when the fatty acid was substituted for (↓); or were adjusted for as a covariate. The MetS was defined using the IDF definition. Fatty acids were entered as percentage of total energy intake. The results are presented as OR for the presence of the MetS per change in the proportion of dietary energy. All models were adjusted for total energy and alcohol intake.

**Table 8 ijerph-19-15452-t008:** Association between dietary macronutrient intake and fatty acids (substitution model) and the components of the MetS (as continuous variables).

	WC		Triglycerides †		HDL-C †		BP †		FBG †	
	β	95%CI	β	95%CI	β	95%CI	β	95%CI	β	95%CI
↑CHO_%E ↓PROTEIN_%E	0.005	−0.001, 0.011	0.000	0.000, 0.001	−0.201	−0.516, 0.114	0.001	−0.005, 0.008	−0.001	−0.005, 0.004
↑CHO_%E ↓FAT_%E	0.006	0.000, 0.012	0.000	0.000, 0.001	−0.149	−0.474, 0.175	0.002	−0.004, 0.008	−0.000	−0.005, 0.004
↑FAT_%E ↓PROTEIN _%E	−0.188	−0.837, 0.460	0.035	−0.016, 0.086	0.008	−0.007, 0.024	0.489	−0.187, 1.165	0.491 *	0.009, 0.972
↑SFA_%E ↓MUFA_%E	0.205	−1.264, 1.673	−0.080	−0.195, 0.034	−0.016	−0.051, 0.019	−0.881	−2.397, 0.636	1.105 *	−2.185, −0.025
↑SFA_%E ↓PUFA_%E	−0.015	−1.060, 1.030	−0.035	−0.113, 0.042	−0.007	−0.030, 0.017	−0.030	−1.127, 1.068	−0.735	−1.509, 0.040
↑MUFA_%E ↓PUFA_%E	0.012	−0.049, 0.072	0.001	−0.003, 0.005	−0.001	−0.002, 0.000	0.006	−0.057, 0.068	−0.030	−0.074, 0.014

SFA—saturated fatty acid; WC—waist circumference; HDL-C—high-density lipoprotein cholesterol; SBP—systolic blood pressure; DBP—diastolic blood pressure; CHO—carbohydrate; MUFA—monounsaturated fatty acid; PUFA—polyunsaturated fatty acid. * *p* < 0.05 All models were adjusted for age, race, physical activity, marital status, level of education, driving experience and smoking status. In each model, a given dietary intake variable of interest was included as an independent variable (↑) and one dietary intake variable (↓) was excluded from the model. The remaining dietary intake variables and total energy were included as covariates. The β represents the increase or decrease in the continuous outcome variable when increasing the independent dietary intake, while simultaneously reducing an isoenergic amount of the excluded dietary intake variable. All dietary variables were entered as percentages of the total energy intake. † Also adjusted for BMI.

## Data Availability

The data presented in this study are available on request from the corresponding author.

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
