# Peer review of "Association between Macronutrient and Fatty Acid Consumption and Metabolic Syndrome: A South African Taxi Driver Survey"

_ijerph, 2022, doi:10.3390/ijerph192315452_

Round 1

Reviewer 1 Report

In this study Authors reported the association between macronutrients and fatty acid intake with metabolic syndrome in male taxi drivers of South Africa. A multivariate nutrient density substitution models was used to analyze the possible substitution of fatty acid and carbohydrates with protein in the diet.

However, as correctly reported by Authors,this is a mathematical model and a real life study is mandatory to assess the importance of diet modification on the development and treatment of metabolic syndrome.

In my opinion it would be very important to quantify the role of lifestyle on the diet because the correct food composition in the diet is essential but the association of high fat and carbohydrates  with  cigarette smoking , excessive alcohol intake  and physical inactivity plays a key role in the development of metabolic syndrome.

I suggest a revision of the manuscript and a  real life comparison with the introduction of 2 or 3 groups of subjects using different lifestyle and diet in order to obtain more interesting results.

Reviewer 2 Report

Dear Authors,

I have reviewed your manuscript entitled "Association of macronutrient and fatty acid consumption with metabolic syndrome: a South African taxi driver survey"

The manuscript is well written and the background of the research is explained in the manuscript. The methodology of data collection is clear, and the results are described concisely and clearly along with the tables. The discussion focuses on the main findings, comparing them to other research conducted, and also states the limitations of your research. The conclusion of the manuscript is clear and points to the need for further research.

What concerns me is that using software to check the authenticity of your manuscript, I found 15% plagiarism, i.e., self-plagiarism, related to your paper published in August this year (https://www.ncbi.nlm.nih.gov/pmc/articles/PMC9406286/). It is reasonable to assume that the results of this manuscript are from the research already mentioned. Therefore, I would suggest that you briefly describe in the introduction the research you have conducted, which you mentioned within reference no [4] (https://www.ncbi.nlm.nih.gov/pmc/articles/PMC9406286/), and indicate why you decided to continue the study.

I believe that in this way you would make a more original contribution to this manuscript.

Reviewer 3 Report

I congratulate the authors for the work developed. They have made a complete review of the topic and have explained it clearly.

I would like to send you some comments related to the work you present.

I consider that focusing on a population as concrete and specific as male mini-bus taxi drivers limits the extrapolation of results to other populations. Has this research been carried out on a larger scale, in another population?

The inclusion criteria have been very restrictive, hence my surprise at having been able to reach that sample of 185 people.

I consider as a limitation that one of the inclusion criteria was eating street food at least 3 times a week. Due to ignorance I ask you the question, is all the street food that is offered unhealthy? I would like clarification on this point.

The variables are well described, as well as the different data collection methods.

The results are well detailed. They have included several tables where these results are reflected. Perhaps the footnotes, in my opinion, include too much text.

Adequate bibliography relevant to the subject of study.

Thanks.

Round 2

Reviewer 1 Report

The manuscript has been partially revised as suggested by the Reviewer.

Reviewer 2 Report

Dear authors,

thank you very much for considering my suggestion to improve your manuscript